

# Image database of Japanese food samples with nutrition information

Wataru Sato[1], Kazusa Minemoto[2], Reiko Sawada[2], Yoshiko Miyazaki[3] and Tohru Fushiki[3]

[1] RIKEN, Kyoto, Japan
[2] Kyoto University, Kyoto, Japan
[3] Ryukoku University, Ohtsu, Japan

## ABSTRACT

**Background:** Visual processing of food plays an important role in controlling eating behaviors. Several studies have developed image databases of food to investigate visual food processing. However, few databases include non-Western foods and objective nutrition information on the foods.

**Methods:** We developed an image database of Japanese food samples that has detailed nutrition information, including calorie, carbohydrate, fat and protein contents. To validate the database, we presented the images, together with Western food images selected from an existing database and had Japanese participants rate their affective (valence, arousal, liking and wanting) and cognitive (naturalness, recognizability and familiarity) appraisals and estimates of nutrition.

**Results:** The results showed that all affective and cognitive appraisals (except arousal) of the Japanese food images were higher than those of Western food. Correlational analyses found positive associations between the objective nutrition information and subjective estimates of the nutrition information, and between the objective calorie/fat content and affective appraisals.

**Conclusions:** These data suggest that by using our image database, researchers can investigate the visual processing of Japanese food and the relationships between objective nutrition information and the psychological/neural processing of food.

Corresponding author
Wataru Sato,
wataru.sato.ya@riken.jp

## INTRODUCTION

Visual processing of food plays an important role in identifying inherent biologically-significant information from food, such as edibility (*Tsourides et al., 2016*). The sight of food elicits affective or hedonic responses (*Rodríguez et al., 2005*) that occur rapidly even before the conscious perception of food (*Sato et al., 2016*), which in turn motivates food intake (*Yeomans, Blundell & Leshem, 2004*). Functional neuroimaging studies have shown that seeing food images and consuming taste solutions activate common neural circuits, including the gustatory and affective brain regions (*Simmons, Martin & Barsalou, 2005*; for reviews, see *Huerta et al., 2014*; *Van Der Laan et al., 2011*).

Studies have created standardized image databases of food to investigate visual food processing (*Foroni et al., 2013*; *Blechert et al., 2014*; *Miccoli et al., 2014*; *Charbonnier et al., 2016*). For example, *Foroni et al. (2013)* collected 252 images of natural or processed food,

43 images of rotten food and 529 images of non-food materials in a web-based search and validated these stimuli by assessing the participants' appraisals, including valence, arousal and familiarity. Their database was used in subsequent studies, such as a functional neuroimaging investigations of visual food processing (*Mengotti, Foroni & Rumiati, 2019*; *Morys, Bode & Horstmann, 2018*; *Padulo et al., 2016*).

However, two issues regarding a food image database remain. First, most of the existing databases contain only Western food. A number of cross-cultural psychological studies have shown that cultural aspects of food modulate hedonic appraisals while seeing and eating food (*Wanich et al., 2018*; *Torrico et al., 2019*; *Sato et al., 2019b*; for a review, see *Prescott et al., 1998*). For example, *Torrico et al. (2019)* showed images of various food products to participants with Western and Asian backgrounds and asked them to rate their preferences. The food images from Western and Asian origins elicited stronger preferences in the Western and Asian groups, respectively. These data suggest the need to develop datasets of non-Western food images. To overcome this problem, a recent study developed a food image database containing 209 images in the Asian food category (*Toet et al., 2019*). It is desirable to develop non-Western food image databases that have different specific advantages to investigate visual food processing further.

Second, the existing databases do not provide nutrition information on the foods. Although the total calories (*Foroni et al., 2013*; *Charbonnier et al., 2016*) or the calories and carbohydrate, fat and protein contents (*Blechert et al., 2014*) in the materials were reported, the information was estimated using general data and not the specific foods in the images (*Foroni et al., 2013*; *Blechert et al., 2014*) or are not described in detail (*Charbonnier et al., 2016*; *Toet et al., 2019*). This issue could be important, because it is widely believed that objective nutrition information affects the visual processing of food (*Birch, 1999*) but empirical data are scarce and mixed. Some studies provided positive evidence that participants accurately estimated the total calories (*Foroni et al., 2013*; *Charbonnier et al., 2016*; *Brunstrom et al., 2018*) and fat content (*Toepel et al., 2009*) and that they showed a preference for high-calorie food (*Brunstrom et al., 2018*) based on visual information on of the food materials. Other studies reported problematic estimates of total calories for food images (*Carels, Konrad & Harper, 2007*; *Foroni, Pergola & Rumiati, 2016*; *Horne et al., 2019*). To investigate these relationships further, objective nutrition information is indispensable for food image stimuli.

To investigate these issues, we developed an image database of Japanese food samples (i.e., plastic replicas) (Fig. 1). We photographed the stimuli selected from a database of Japanese food samples that looked realistic, included Japanese food that frequently appears in contemporary Japanese home meals, and have detailed nutrition information (e.g., calorie, carbohydrate, fat and protein contents). To validate the image database, we assessed the participants' subjective appraisals of their affective (valence, arousal, liking and wanting) appraisals of the images. We also assessed cognitive appraisals to confirm whether food images looked as natural as food (naturalness) and were recognizable (recognizability) and familiar (familiarity). As reference data in relation to these appraisals, we additionally included images of Western food selected from an existing database (*Blechert et al., 2014*). We also assessed the subjective estimate of nutrition and investigated

the relationships between objective nutrition information and the subjective perception of nutrition, as well as affective appraisals.

# MATERIALS AND METHODS

## Participants

The appraisal experiment enrolled 32 Japanese volunteers (13 females; mean ± SD age, 21.9 ± 3.4 years). The required sample size was determined based on an a priori power analysis using G*Power software ver. 3.1.9.2 (*Faul et al., 2007*). The *d* of 0.5 (medium-size effect), an α level of 0.05, and a power (1−β) of 0.80 were assumed in analyses of relationships between objective nutrition information and subjective appraisals. The result of the power analysis showed that more than 27 participants were needed. Participants were gathered through advertising presented at the Kyoto University facility, and participants received a book coupon corresponding to 500 Japanese yen. All participants were confirmed not to have any food restrictions for medical or religious reasons. The participants' hunger levels were rated on a 9-point scale from 1 (very hungry) to 9 (very satiated) before the experiment, and the results showed that the majority of them were in the neutral state (mean ± SD, 4.5 ± 1.6). None of the participants were obese (body mass index, <30, mean ± SD, 20.4 ± 2.5 kg/m$^2$). All participants had normal or corrected-to-normal visual acuity and had no color vision deficiencies. After a detailed explanation of the experimental procedure, all participants provided written informed consent. This study was approved by the Ethics Committee of the Graduate School of Medicine, Kyoto University (R0343), and was conducted in accordance with approved guidelines.

## Apparatus

The experimental events were controlled by PowerPoint 2007 (Microsoft, Redmond, WA, USA) implemented on a laptop computer (Precision M6300, Dell, Round Rock, TX, USA).

## Stimuli

As Japanese food stimuli, we photographed items in a Japanese food sample database, the Syokuiku Satisfactory "à La Carte" Tray (SAT) system (Iwasaki, Osaka, Japan; http://www.foodmodel.com/category12/index.html). "Syokuiku" is a Japanese term that means the education/promotion of food and nutrition. The database contains 118 samples of Japanese foods that frequently appear in contemporary Japanese home meals (e.g., meat and potato stew). The food samples bear a close resemblance to actual food. Importantly, the SAT system has nutrient information for each food item, including weight (g), calories (kcal), moisture (mL), carbohydrate (g), fat (g) and protein (g), based on a detailed analysis of materials included in the specific item (e.g., 7.5 g light soy sauce in #07 Kitsune udon). We selected 46 processed food items (i.e., removing non-processed food such as fruits) without obstacles (i.e., packaging and plastic covers) and took color pictures using a digital camera (EXILIM FH100; Casio, Tokyo, Japan). We selected only processed food items, because the SAT set had a relatively small number of non-processed food items, and previous studies have shown that affective appraisals can be different across processed vs.

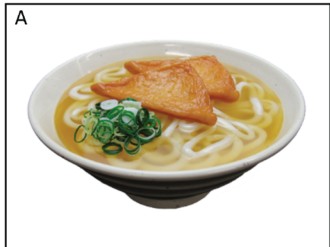
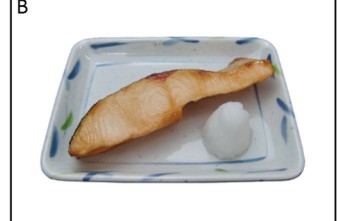
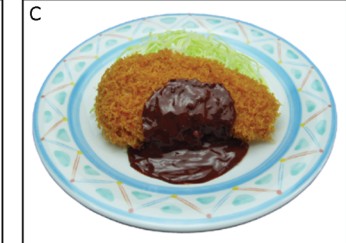

**Figure 1 Examples of food stimuli.** The items #07 (Kitsune udon; A), #10 (Sake no shioyaki; B), and #26 (Tonkatsu; C) are shown.

non-processed food (*Coricelli et al., 2019a*). The pictures were then cropped and modified to remove the background using Photoshop CS6 (Adobe, San Jose, CA, USA).

For Western food stimuli, we selected five pictures of Western processed food (e.g., meatballs; #190, 315, 322, 324 and 384) from the existing database (*Blechert et al., 2014*). We selected only five images because these were used as the reference condition with the validated database images of actual food. We did not intend to compare thoroughly Western vs. Japanese food images. Previous methodological studies suggested that three items would be needed to represent a group reliably (*MacCallum et al., 1999*; *Raubenheimer, 2004*).

All stimuli measured horizontally 600 × vertically 450 pixels. Some examples of the Japanese food stimuli are shown in Fig. 1 and all are shown in Fig. S1. The pictorialized Western food images are shown in Fig. S2.

## Procedure

To validate the image database of the Japanese food samples, we conducted a appraisal experiment using the images of Japanese and Western food with a paper–pencil questionnaire. The participants were tested individually. The images were presented one at a time. Participants advanced the image by themselves after appraising all items and they were forbidden to return to any item. For each image, the participants were asked to rate valance (from "negative" to "positive"), arousal (from "low arousal" to "high arousal"), and liking, wanting, naturalness, recognizability and familiarity (from "not at all" to "very much") on 9-point Likert scales. The participants were also asked to estimate the total calories (kcal) and caloric percentages of carbohydrate, fat and protein and describe the exact figures. In total, 51 trials (46 for Japanese food; five for Western food) were performed. The order of trials was randomized. At the beginning of the experiment, thumbnails of all images were presented to enable participants to perceive all the images and rate them using a wide range of scales (i.e., to reduce the anchor effect in Likert scales; *Bishop & Herron, 2015*), and then two practice trials were conducted using images not included in the database.

## Data analysis

All statistical tests were performed using SPSS 16.0J software (SPSS Japan, Tokyo, Japan). First, to compare the appraisals of Japanese and Western food images, we calculated the

mean appraisal for each image across participants. Welch's *t*-tests (two-tailed) were conducted for appraisals (valence, arousal, liking, wanting, naturalness, recognizability and familiarity); Welch's *t*-tests are more robust to unequal sample size (*Delacre, Lakens & Leys, 2017*). The effect size *r* (*Cohen, 1992*) was calculated.

Next, to evaluate the relationships between objective nutrition information (calories (kcal) and relative caloric percentages of carbohydrate, fat and protein) and subjective nutrition/affective appraisals for the Japanese food images, we calculated the Pearson's product-moment correlation coefficient between the objective nutrition information and subjective appraisals across images for each participant. Then, the correlation coefficients were normalized using a Fisher's *r*-to-*z* transformation and entered into Student's one-sample *t*-tests to evaluate a significant difference from zero (two-tailed). We used this two-stage random effect analyses to evaluate the generalizability of individual-level statistical models, rather than calculating the correlations between aggregated data (cf. *Zhang & Wang, 2014*). However, to visualize heuristically the relationships between objective nutrition information and subjective appraisals, we depicted the scatterplots and regression lines using the aggregated (group-mean) data in supplementary figures. In addition, we analyzed the relationships between the subjective estimates of nutrition information and affective appraisals in the same way for descriptive purposes. The results were considered statistically significant at $p < 0.05$.

We conducted preliminary analyses of the factor sex and found no significant effects on the results. Hence, this factor was disregarded. We also performed preliminary analyses of the visual properties (brightness, spatial frequency (i.e., overall activity level; *Eskicioglu & Fisher, 1995*; *Li, Kwok & Wang, 2001*), and entropy (i.e., measure of randomness; *Tsai, Lee & Matsuyama, 2008*) of each Japanese food sample image using MATLAB 2018 (MathWorks, Natick, MA, USA). We confirmed that almost all of the reported significant associations between the objective nutrition information and subjective appraisals were significant even when the covariates (cf. *Hedberg & Ayers, 2015*) of the correlation coefficients between visual properties and subjective appraisals were included (File S1).

## RESULTS

### Information of images

Figure 1 shows examples of images of the Japanese food samples and Table 1 lists the names and descriptions of all 46 items. Table 2 shows the mean (with SD) subjective appraisals across all Japanese food samples. The mean (with SD) objective nutrition information of Japanese food samples came to: total calories, 236.7 ± 194.1 kcal; %carbohydrate, 34.99 ± 25.6; %fat, 38.1 ± 20.8; and %protein, 26.8 ± 17.2.

### Subjective affective and cognitive appraisals

Figure 2 shows the mean (with SE) subjective affective (valence, arousal, liking and wanting) and cognitive (naturalness, recognizability and familiarity) appraisals for each item. Welch's *t*-tests contrasting Japanese vs. Western food images showed significant differences for all measures ($t(49) > 5.52$, $p < 0.001$, $r > 0.61$; Table 3), indicating higher

**Table 1 Names and descriptions of the food items.**

| ID | Name | Description |
|---|---|---|
| Japanese | | |
| 01 | Oyakodon | A bowl of rice with chicken, egg, and vegetables |
| 02 | Aij no hiraki | Grilled sun-dried horse mackerel |
| 03 | Tempura moriawase | Deep-fried fish and vegetables in a light batter |
| 04 | Nikujaga | Meat and potato stew |
| 05 | Chikuzenni | Chicken stew with vegetables |
| 06 | Buri no teriyaki | Grilled teriyaki flavored yellowtail |
| 07 | Kitsune udon | Udon noodles with deep-fried tofu |
| 08 | Nigiri sushi | Hand-formed sushi |
| 09 | Hiyayakko | Cold tofu |
| 10 | Sake no shioyaki | Grilled salmon with salt |
| 11 | Natto | Fermented soybeans |
| 12 | Dashimaki | Rolled Japanese-style omelet |
| 13 | Saba no nitsuke | Simmered mackerel |
| 14 | Kabocha no nimono | Boiled pumpkin |
| 15 | Yasai no nimono | Boiled vegetables |
| 16 | U no hana | Bean curd dregs |
| 17 | Gomoku nimame | Boiled beans |
| 18 | Komatsuna no ohitashi | Soaked Japanese mustard spinach |
| 19 | Kimpira gobo | Kimpira-style sautéed burdock |
| 20 | Kiribothi daikon no nimono | Stewed dried radish |
| 21 | Sirloin steak | Sirloin steak |
| 22 | Buta no shogayaki | Ginger-fried pork |
| 23 | Tori no karaage | Fried chicken |
| 24 | Wakame to kyuri no tsukemono | Vinegared Wakame seaweed and cucumber |
| 25 | Mix fry | Assorted breaded deep-fried food |
| 26 | Tonkatsu | Pork cutlet |
| 27 | Butanikuiri yasaiitame | Fried vegetables with pork |
| 28 | Hamburg | Hamburg steak |
| 29 | Katsuo no tataki | Lightly roasted bonito |
| 30 | Spaghetti meat sauce | Spaghetti with meat sauce |
| 31 | Curry and rice | Curry and rice |
| 32 | Sanma no shioyaki | Saury grilling fish with salt |
| 33 | Plain omelet | Plain omelet |
| 34 | Sashimi moriawase | Assorted sliced raw fish |
| 35 | Takoyaki | Octopus dumplings |
| 36 | Set yakimeshi | Fried rice |
| 37 | Gyoza | Gyoza dumplings |
| 38 | Chawanmushi | A savory steamed egg custard with assorted ingredients |
| 39 | Koyadohu no nimono | Boiled freeze-dried tofu |
| 40 | Oden | Vegetables, fish dumplings and various other stewed items |

| ID | Name | Description |
|----|------|-------------|
| 41 | Tamagodofu | Steamed egg custard |
| 42 | Yasai no misoshiru | Miso soup with vegetables |
| 43 | Asari no sumashijiru | Clear soup with clams |
| 44 | Anpan | A sweet roll filled with red bean paste |
| 45 | Potate fri | French fried potatoes |
| 46 | Doughnut | A donut |
| Western | | |
| 01 | Frikadellen | Flat, pan-fried meatballs of minced meat (#190) |
| 02 | Shashlik | Skewered and grilled cubes of meat (#315) |
| 03 | Knuckle of pork with sauerkraut | Leg of pork with sauerkraut (#322) |
| 04 | Rissole | A small fried ball of chopped meat or vegetables (#324) |
| 05 | Tortellini | Ring-shaped pasta (#384) |

Note:
The names of the Japanese food are in Japanese. Numbers with the Western food descriptions refer to the item numbers in the database (Blechert et al., 2014).

**Table 2 Mean (with standard deviation) subjective appraisals for all Japanese food sample images.**

| Valence | Arousal | Liking | Wanting | Naturalness | Recognizability | Familiarity | Calorie | Carbohydrate | Fat | Protein |
|---------|---------|--------|---------|-------------|-----------------|-------------|---------|--------------|-----|---------|
| 5.9 | 5.5 | 6.1 | 5.5 | 7.7 | 5.6 | 5.9 | 242.2 | 33.1 | 18.9 | 48.0 |
| (0.6) | (0.9) | (0.8) | (0.8) | (0.8) | (0.8) | (1.0) | (127.7) | (16.8) | (8.5) | (17.3) |

Note:
Affective and cognitive appraisals (valence–familiarity) are in 9-point scale in 1–9. Calorie is in kcal, and carbohydrate, fat, protein are in %.

appraisals for Japanese foods than for Western foods, except the arousal appraisal ($t(49) = 1.00$, $p > 0.10$, $r = 0.14$). The data suggest that the images of Japanese food samples in the database elicited stronger food-related positive affective appraisals (positive, liking and wanting), and looked more natural, recognizable, and familiar than the photographs of Western food selected from the existing database (Blechert et al., 2014).

## Relationships between objective nutrition information and subjective food appraisals

Next, we analyzed the relationships between the objective nutrition information (calories, carbohydrate, fat and protein) and subjective nutrition/affective appraisals for food. Figure 3 shows the correlation coefficients between the objective information and subjective estimates of nutrition (cf. group-mean scatterplots with regression lines in Fig. S3). Student's one-sample $t$-tests after Fisher's $r$-to-$z$ transformation showed that all of the relationships were positive and significant ($t(31) > 11.32$, $p < 0.001$, $r > 0.53$; Table 4).

Figure 4 shows the correlation coefficients between the objective information and affective appraisals (cf. group-mean scatterplots with regression lines in Fig. S4). Student's one-sample $t$-tests after Fisher's $r$-to-$z$ transformation for objective information showed that objective calories and fat were significantly and positively correlated with all of the affective food appraisals ($t(31) > 2.70$, $p < 0.05$, $r > 0.37$; Table 5). Significant negative

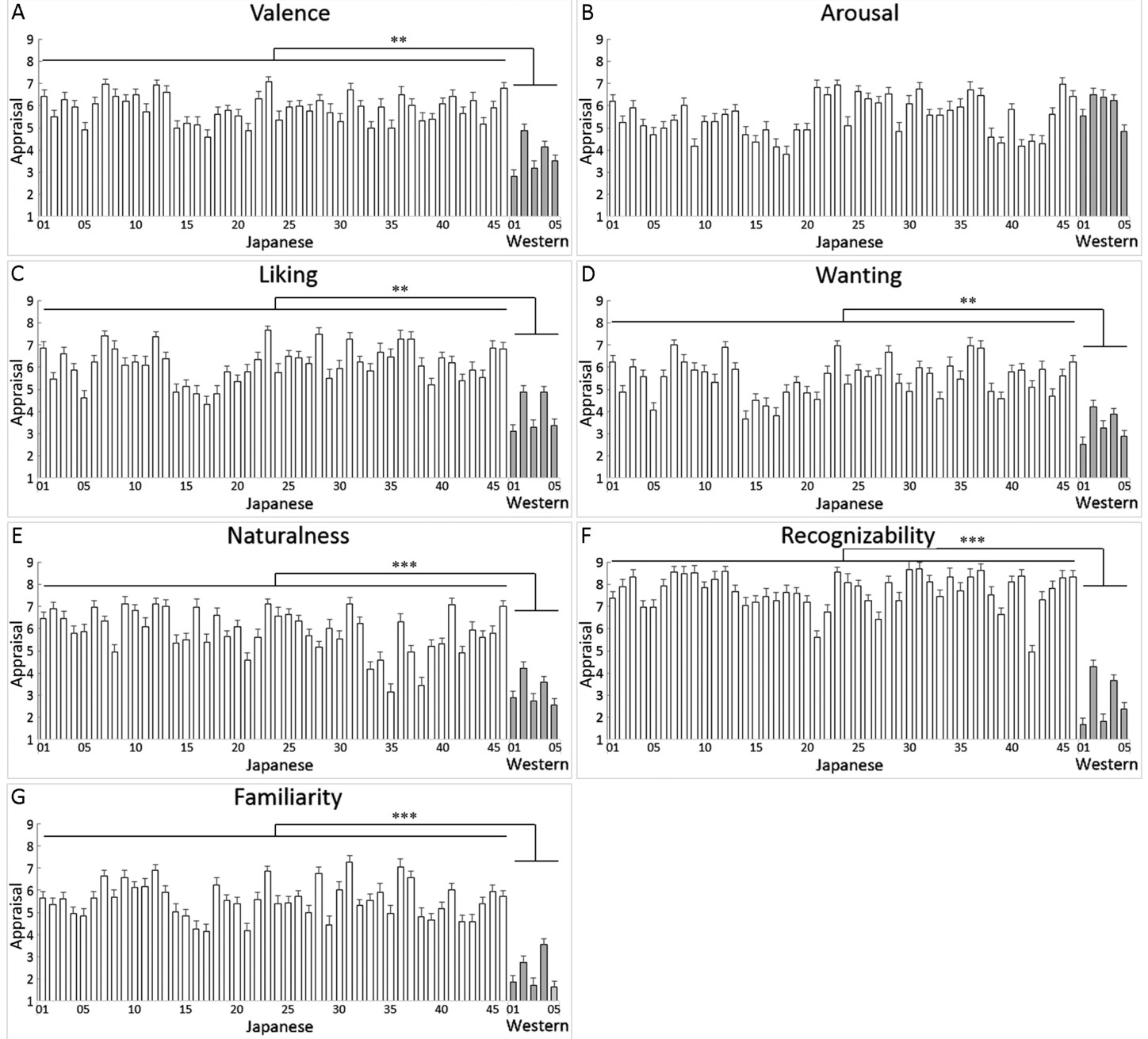

**Figure 2 Mean (with standard error) subjective ratings of affective and cognitive appraisals of Japanese food sample images and Western food images.** Numbers on the *x*-axes are the items in Table 1. ***, *p* < 0.001 (*t*-tests). Affective (valence (A), arousal (B), liking (C), and wanting (D)) and cognitive (naturalness (E), recognizability (F) and familiarity (G)) appraisals are shown. Numbers on the *x*-axes are the items in Table 1. ***, *p* < 0.001 (*t*-tests).

correlations were detected between objective carbohydrate and valence, liking, and wanting (*t*(31) > 2.72, *p* < 0.05, *r* > 0.42) and between objective protein and arousal (*t*(31) = 6.02, *p* < 0.001, *r* = 0.69).

In addition, we exploratorily analyzed the relationships between the subjective estimates of nutrition information and affective appraisals (Fig. 4). The analysis showed the same

**Table 3 Results of Welch's *t*-tests (two-tailed) contrasting subjective appraisals for Japanese vs. Western food images.**

| Statistic | Valence | Arousal | Liking | Wanting | Naturalness | Recognizability | Familiarity |
|---|---|---|---|---|---|---|---|
| *t* | **5.76** | 1.21 | **5.35** | **6.45** | **9.29** | **8.34** | **7.95** |
| *p* | **0.003** | 0.277 | **0.004** | **0.001** | **0.000** | **0.000** | **0.000** |
| *r* | **0.94** | 0.46 | **0.93** | **0.94** | **0.98** | **0.97** | **0.96** |

Note:
Degrees of freedom were 4.42–5.87. Significant results ($p < 0.05$) are in bold.

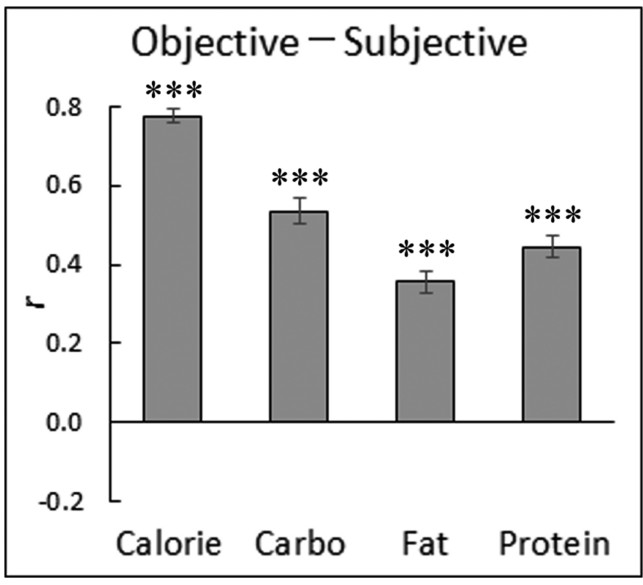

**Figure 3 Mean (with standard error) intra-individual correlation coefficients between objective and subjective nutrition information for Japanese food sample images.** ***, $p < 0.001$ (*t*-tests).

**Table 4 Results of Student's one-sample *t*-tests (two-tailed) for the correlation coefficients between the objective information and subjective estimation of nutrition.**

| Statistic | Calories | Carbohydrate | Fat | Protein |
|---|---|---|---|---|
| *t* | **25.62** | **12.72** | **11.32** | **15.04** |
| *p* | **0.000** | **0.000** | **0.000** | **0.000** |
| *r* | **0.98** | **0.91** | **0.89** | **0.93** |

Note:
Data were analyzed after Fisher's *r*-to-*z* transformation. Degrees of freedom were 31. Significant results ($p < 0.05$) are in bold.

significant patterns as the results of objective information for subjective calories and fat ($t(31) > 2.70$, $p < 0.05$, $r > 0.37$; Table 5), except that the subjective fat–valence relationship did not reach significance ($t(31) = 1.57$, $p > 0.10$, $r = 0.26$). Subjective carbohydrate and protein showed rather different patterns, including a significant positive correlation between subjective carbohydrate and arousal ($t(31) = 2.27$, $p < 0.05$, $r = 0.36$) and negative correlations between subjective protein and arousal, liking, and wanting ($t(31) > 2.09$, $p < 0.05$, $r > 0.33$).

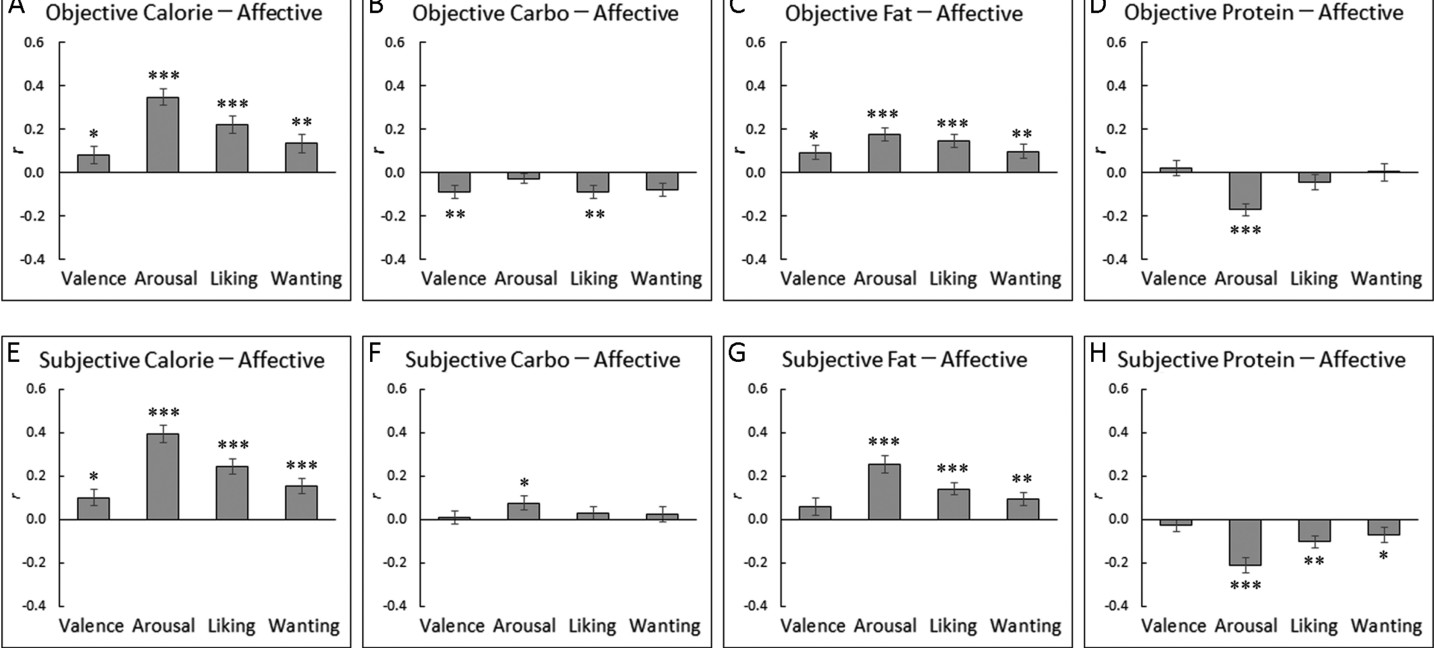

**Figure 4 Mean (with standard error) intra-individual correlation coefficients between objective/subjective nutrition information and affective ratings for Japanese food sample images.** The correlations of affective ratings with objective calorie (A), objective carbohydrate (B), objective fat (C), objective protein (D), subjective calorie (E), subjective carbohydrate (F), subjective fat (G), and subjective protein (H) are shown. ***, $p < 0.001$; **, $p < 0.01$; *, $p < 0.05$ (one-sample $t$-tests after Fisher's transformation).

## DISCUSSION

Our data on the subjective affective appraisals showed that the images of Japanese food samples were more positive, liked, and wanted by Japanese participants than the five selected images of Western food items. The naturalness appraisals showed that the images of Japanese food samples were rated more natural than the Western food images selected from the existing database. This result suggests that the food sample images looked almost as real as actual food. The results for familiarity and recognizability showed that the Japanese foods in the database were more familiar and more easily recognizable by Japanese participants than the selected Western food items. Overall, these results indicate that the images of Japanese food samples we developed validly represent Japanese foods.

Because our food image database had objective nutrition information, we analyzed the relationships between objective nutrient information and subjective nutrition and affective appraisals. The results showed that the objective information and subjective perception of nutrition were positively associated, and that the objective caloric content and percentage of fat calories were positively associated with all of the affective appraisals. These results are consistent with previous findings that participants estimated the caloric (*Foroni et al., 2013*) and fat (*Toepel et al., 2009*) contents precisely and reported a greater liking for food with high caloric values (*Brunstrom et al., 2018*) during visual food processing, although other studies failed to find such patterns (*Carels, Konrad & Harper, 2007*) and debate remains. However, previous studies estimated the nutrition information using a general database and did not investigate the contents of specific food

**Table 5 Results of Student's one-sample *t*-tests (two-tailed) for the correlation coefficients between the objective/subjective nutrition and affective appraisals.**

| Nutrition | Statistic | Affective appraisal | | | |
|---|---|---|---|---|---|
| | | Valence | Arousal | Liking | Wanting |
| Objective calorie | *t* | **2.34** | **9.28** | **6.02** | **3.46** |
| | *p* | **0.025** | **0.000** | **0.000** | **0.001** |
| | *r* | **0.37** | **0.85** | **0.72** | **0.51** |
| Objective carbohydrate | *t* | **2.62** | 0.98 | **2.67** | **2.28** |
| | *p* | **0.013** | 0.336 | **0.012** | **0.029** |
| | *r* | **0.41** | 0.16 | **0.42** | **0.36** |
| Objective fat | *t* | **2.99** | **6.03** | **5.05** | **3.29** |
| | *p* | **0.005** | **0.000** | **0.000** | **0.002** |
| | *r* | **0.46** | **0.72** | **0.65** | **0.49** |
| Objective protein | *t* | 0.84 | **5.53** | −1.11 | 0.27 |
| | *p* | 0.409 | **0.000** | 0.274 | 0.793 |
| | *r* | 0.14 | **0.69** | 0.19 | 0.05 |
| Subjective calories | *t* | **2.73** | **9.03** | **6.73** | **4.29** |
| | *p* | **0.010** | **0.000** | **0.000** | **0.000** |
| | *r* | **0.42** | **0.84** | **0.76** | **0.59** |
| Subjective carbohydrate | *t* | 0.26 | **2.27** | 1.03 | 0.75 |
| | *p* | 0.799 | **0.030** | 0.313 | 0.459 |
| | *r* | 0.04 | **0.36** | 0.17 | 0.13 |
| Subjective fat | *t* | 1.57 | **6.06** | **4.80** | **3.24** |
| | *p* | 0.127 | **0.000** | **0.000** | **0.003** |
| | *r* | 0.26 | **0.72** | **0.64** | **0.49** |
| Subjective protein | *t* | 0.96 | **5.98** | **3.44** | **2.10** |
| | *p* | 0.342 | **0.000** | **0.002** | **0.044** |
| | *r* | 0.16 | **0.72** | **0.51** | **0.34** |

Note:
Data were analyzed after Fisher's *r*-to-*z* transformation. Degrees of freedom were 31. Significant results ($p < 0.05$) are in bold.

images. The previous studies also did not investigate systematically the carbohydrate, fat, and protein contents. Therefore, we investigated this and found that the relative caloric ratio of carbohydrate and protein was negatively associated with the affective appraisals of food. Our results also showed that the relationships between objective nutrition information and affective appraisals are rather similar to those between subjective nutrition perception and affective appraisals in terms of caloric and fat contents, but not in terms of carbohydrate and protein. In summary, our results confirm and extend previous findings indicating that objective nutrition information about food can influence the subjective estimates of nutrition and affective appraisals during the visual processing of food.

The database of Japanese food images with nutrition information that we developed has practical significance for research on visual food processing. For example, it would be

interesting to use our database in functional neuroimaging studies. Previous studies have shown that several brain regions, including the visual cortices (e.g., the fusiform gyrus) and limbic regions (e.g., the amygdala), are activated more during the presentation of food images than non-food images (*Holsen et al., 2005*; *Sato et al., 2019a*; for a review, see *Van Meer et al., 2015*). Although a few studies compared neural activity in response to low vs high calorie/fat food (*Frank et al., 2010*; *Killgore et al., 2003*; *Toepel et al., 2009*), more detailed parametric relationships between neural activity and objective nutrition information remain unexplored. Use of the current database may allow studies of the specific brain activities associated with objective nutrition.

Several limitations of this study should be acknowledged. First, our database was restricted to a small number of food samples. Therefore, the number of images should be increased. Developing a comparable image database using real food with detailed nutrition information may also be helpful. Second, our stimuli were restricted to processed food. Because some previous studies have suggested different psychological (*Aiello et al., 2018*; *Coricelli et al., 2019a*; *Rumiati et al., 2016*) and neural (*Coricelli et al., 2019b*; *Pergola et al., 2017*) processing between processed and non-processed food, the generalizability of the current results for non-processed food is an important matter for future research. Third, our food stimuli were depicted in different plates. This was because the Japanese diet has custom rules for plates (*Thompson, 2016*) and the food sample set we photographed mimicked this to create realistic Japanese food stimuli. This may be problematic, because some previous studies have shown that plates can influence the affective response to food items (*Piqueras-Fiszman et al., 2012*; *Piqueras-Fiszman, Giboreau & Spence, 2013*; *Stewart & Goss, 2013*; *Van Ittersum & Wansink, 2012*). An assessment using the present images after removing the plates is warranted to confirm the findings. Fourth, we assessed only subjective appraisals. Because subjective appraisals could be biased to the results that researchers want to find due to participants' care about demand characteristics (*Orne, 1962*), objective measures of affective responses, such as physiological signals (*Kaneko et al., 2018*), may complement the current findings. Finally, we tested only a small sample of participants. We determined the sample size to detect more than middle size effects (cf. *Cohen, 1992*). Because our sample included only young participants, it remains unknown whether participants of different age groups, such as children and older participants, would show similar affective and cognitive appraisals, and concordance between objective nutrition information and subjective appraisals. We tested only Japanese participants; hence, the patterns may be different in different cultures. We also did not assess the details of participants' characteristics that could modulate food processing, such as dieting habits (*Coricelli et al., 2019a*; *Hoefling & Strack, 2008*). Investigations including more participants from different age groups and different cultures, with detailed assessments of their characteristics, by using the present stimulus database would be valuable for investigating visual food processing further.

## CONCLUSIONS

We developed an image database of Japanese food samples that contains detailed nutrition information, including calorie, carbohydrate, fat and protein contents. The appraisal

experiments showed that all affective and cognitive appraisals of the Japanese food sample images were higher than, or comparable to, those of Western foods. Correlational analyses showed positive associations between the objective information and subjective perception of the calorie, carbohydrate, fat and protein contents, and objective calorie/fat content and affective appraisals. These data suggest that by using our image database, researchers can investigate the visual processing of Japanese food and relationships between objective nutrition information and psychological/neural processing of food. This image database is available on request from the corresponding author solely for research purposes. The nutrition information for the stimuli is available from the Iwasaki SAT system (http://www.foodmodel.com/category12/index.html).

## ACKNOWLEDGEMENTS
The authors would like to thank Dr. Hiromi Hata and Ms. Yukari Sato for their technical support.

### Funding
This study was supported by grants from the Project of the NARO Bio-oriented Technology Research Advancement Institution (Integration Research for Agriculture and Interdisciplinary Fields, Japan), the Research Complex Program from Japan Science and Technology Agency, and Japan Society for the Promotion of Science KAKENHI (18K03174). The funders had no role in study design, data collection and analysis, decision to publish, or preparation of the manuscript.

### Grant Disclosures
The following grant information was disclosed by the authors:
NARO Bio-oriented Technology Research Advancement Institution (Integration Research for Agriculture and Interdisciplinary Fields, Japan).
Research Complex Program from Japan Science and Technology Agency, and Japan Society for the Promotion of Science KAKENHI: 18K03174.

### Competing Interests
The authors declare that they have no competing interests.

### Author Contributions
- Wataru Sato conceived and designed the experiments, performed the experiments, analyzed the data, prepared figures and/or tables, authored or reviewed drafts of the paper, and approved the final draft.
- Kazusa Minemoto performed the experiments, authored or reviewed drafts of the paper, and approved the final draft.
- Reiko Sawada performed the experiments, authored or reviewed drafts of the paper, and approved the final draft.

- Yoshiko Miyazaki performed the experiments, authored or reviewed drafts of the paper, and approved the final draft.
- Tohru Fushiki conceived and designed the experiments, performed the experiments, authored or reviewed drafts of the paper, and approved the final draft.

## Human Ethics

The following information was supplied relating to ethical approvals (i.e., approving body and any reference numbers):

This study was approved by the Ethics Committee of the Graduate School of Medicine, Kyoto University (R0343), and was conducted in accordance with approved guidelines.

## Data Availability

The raw measurements, including the representative images used in this analysis, are available as a Supplemental File.

Additional images from the complete image database are available on request from the corresponding author solely for research purposes.

## Supplemental Information

Supplemental information for this article can be found online at http://dx.doi.org/10.7717/peerj.9206#supplemental-information.

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
