# Peer review of "Image database of Japanese food samples with nutrition information"

_PeerJ, doi:10.7717/peerj.9206_

## Round 0.1 · original submission · Major Revisions

· Academic Editor

Major Revisions

Please consider all the modifications suggested by the Reviewer. In particular, Reviewer 1 highlighted a very importante aspect, i. e., the fact that the database seems to be available only from buyers of a particular system. Please explain this properly, together with the potential conflicts of interest derived from it.

·

Basic reporting

This paper presents a validated image database of Japanese food items. The availability of such a database is important for cross-cultural food research. The paper is generally ell written and easy to follow.

An essential reference to an existing and freely available validate food image database that also contains many Japanese and Thai images is missing.

The raw data only contains the demographics and rating data. The food image database itself is not publicly available and will only be supplied to buyers of a “Iwasaki SAT system”, which appears to be a set of plastic food models.

Experimental design

Some details about the experimental designs are missing.

Validity of the findings

The restricted age range (only young people) and the small number of food items (only 46). clearly limits the general validity of the results.

The variation in plateware is a serious limitation (especially when the images are to be used in fMRI studies) that should be mentioned.

Additional comments

General comments

My main concerns are (1) the fact that the photographs appear to represent replicas of plastic food items, which raises doubts about their naturalness, and (2) the fact that the image database is not made publicly available.

Abstract

Line 26: “To verify the validity of the database”: verifying implies that the validity of the database has already been established, which is not the case here. This should be: “To validate the database”.

Line 28 and rest of the text: liking and wanting are not emotional responses, but affective appraisals

Line 32: “better” is incorrect, you probably mean that the ratings were higher

line 34: “and objective information and subjective perception of …” >> “and between objective nutrition information and subjective perception of …” >>

keywords: add valence and arousal

1. Introduction

Line 60: “existing databases contain only Western food” : this is incorrect, for instance, the publicly available CROCUFID food image database contains images of over 200 Japanese food items and more than 100 Thai food items (see Toet, et al., 2019, available at https://osf.io/5jtqx/).

2.1 Participants
Some essential information is missing, like whether the participants had any color vision deficiencies, whether they were following diets or whether they had any food allergies, and their current physical state (hungry, thirsty).
Did the participants receive a financial compensation?
How were the participants recruited? From their low average age I assume that they were all students. How representative is this sample for the general Japanese population? Young people may be more familiar with Westerns food, so the difference may become more significant when a wider age range is included.

2.3 Stimuli

Lines 109-110: I understand (not completely sure since the web page to which the authors refer is only available in Japanese) that the Iwasaki food items are plastic replicas of real food. This is a serious limitation: how realistic are these images compared to real food? Why did the authors did not photograph real food items?

Line 111: The link refers to a site in Japanese; I was therefore unable to verify what it is exactly about. Please explain what the Iwasaki SAT (what does SAT stand for?) system is and why the reader should buy it to obtain the image database (line 253).

Lines 113-114: “The food samples bear a close resemblance to actual food.” >> Please explain what these “food samples” are when they are not real food but when they only resemble food? Why did the authors not make pictures of real Japanese food?

Line 121: Please provide the identifiers of the 5 images from the Food-pics database that were selected for this study. Why did this study include only 5 Western food images?

2.4 Procedure
Was the stimulus presentation self-paced or timed?
What was the reason to show thumbnails of all images at the start of the experiment?

3.1 Image information

Lines 113-114: “Figure 1 shows examples of images of the Japanese food samples” : it appears that all food items were plated differently. It is known that plateware significantly affects emotional and affective response to food items (Piqueras-Fiszman, et al., 2012, Piqueras-Fiszman, et al., 2013, Stewart & Goss, 2013, Van Ittersum & Wansink, 2012). This may be a confounding factor in the present study, especially since the items from the Food-pics database are not plated. Most existing food image databases therefore use a uniform or constant background. Using different backgrounds for each food item makes the images useless for brain studies, since one cannot be sure that differences in brain activity are related to differences in food characteristics or in background characteristics.

Line 159: “Table 1 lists the names and descriptions of the food” >> please mention that this table lists all 46 items in the database

3.2 Subjective ratings
Line 166: 7 t-tests were performed, but only one (which?) result is reported for the significant differences. Also, please add the effect size.

4. Discussion

Line 195: “the images of Japanese food samples were more natural” >> Please rephrase as “the images of Japanese food samples were rated more natural”

Line 199: “than Western foods” >> This is stated too general. Please rephrase, e.g. as “than the five selected Western food items”.

Line 223: The practical significance of this database cwould increase if it was made publicly available. Now it is only offered to buyers of the Iwasaki SAT system

Line 234: The variation in plateware is a serious limitation (especially when the images are to be used in fMRI studies) that should be mentioned.

Other limitations are the restricted age range (young people), which clearly limits the general validity of the results, and the small number of food images (only 46).

5. Conclusion
Line 245: “better” is incorrect, you probably mean that the ratings were higher


Raw data:
“Fe1ale” should be “Female”


References

Piqueras-Fiszman, B., Alcaide, J., Roura, E., & Spence, C. (2012). Is it the plate or is it the food? Assessing the influence of the color (black or white) and shape of the plate on the perception of the food placed on it. Food Quality and Preference, 24 (1), 205-208. doi: 10.1016/j.foodqual.2011.08.011.
Piqueras-Fiszman, B., Giboreau, A., & Spence, C. (2013). Assessing the influence of the color of the plate on the perception of a complex food in a restaurant setting. Flavour, 2 (1), 24. doi: 10.1186/2044-7248-2-24.
Stewart, P.C., & Goss, E. (2013). Plate shape and colour interact to influence taste and quality judgments. Flavour, 2 (1), 27. doi: 10.1186/2044-7248-2-27.
Toet, A., Kaneko, D., de Kruijf, I., Ushiama, S., van Schaik, M.G., Brouwer, A.-M., Kallen, V., & van Erp, J.B.F. (2019). CROCUFID: A cross-cultural food image database for research on food elicited affective responses. Frontiers in Psychology, 10 (58). doi: 10.3389/fpsyg.2019.00058.
Van Ittersum, K., & Wansink, B. (2012). Plate size and color suggestibility: The Delboeuf Illusion’s bias on serving and eating behavior. Journal of Consumer Research, 39 (2), 215-228. doi: 10.1086/662615.

·

Basic reporting

1. English:
Overall, the English language used in the manuscript is clear, however, changes must be done in order to improve the readability of the manuscript to an international audience. In particular, the Abstract section (lines 20-39) requires major improvements in English language and concepts presentation. Stimuli section (lines 109-123) also needs major improvements since it is not clear as it is presented.

2. Literature references:
- In the Introduction section the role of visual processing in food perception is not correctly referenced (lines 43-49 in particular), important papers present in the literature are missing, such as Van der Laan et al., (2011; NeuroImage) and Huerta et al., (2014; Obesity), important meta-analysis works for the field.

- In line 48 the authors refer to “seeing annd eating food” here they should specify that seeing refers to pictorial representations (images) of food and eating refers to liquid tastants stimuli (see Huerta et al., 2014, Obesity).
Moreover, when referring to neuroimaging studies which used the abovementioned pictures databases the authors refer to a single paper, more papers must be cited here (line 58).

- The authors omitted a whole line of research on processed and unprocessed food when describing the stimuli (only processed foods in this study lines 116-117), previous findings using different techniques clearly find differences, both at the behavioral level and neural substrates, in response to these stimuli (see Aiello et al., 2018, Cortex; Coricelli et al., 2019, European Journal Neuroscience).

- When describing results on calories/fat estimations by visual inspection of stimuli (lines 74-81) the authors do not mention an important result by Horne et al. (2019, Appetite) which find that most participants were unable to accurately estimate the caloric content of most of the presented foods (images that were selected from FoodPics, FRIDa and OLAF databases). This point is relevant in light of the findings of the present manuscript (Figure 4 and S2).

3. Structure, Figures, Tables:
- Figure 1 description should contain the description of the presented stimuli, exemplar stimuli of the Western foods should be included in Figure 1

- The complete set of stimuli should be provided as Supplemental Material

- Figure 2 the dimensions of the labels of the axes (too small) and the quality of the image (bad) must be fixed in the revised version of the manuscript

- Quality of images in Figure 3 and Figure 4 is also not optimal and should be improved

- Table 1 should be revised in the description of Western foods; English description should be added (German description can be mantained but in brackets)

- Supplementary Figure 1 (S1) regression values and p values should be added to each plot

- Supplementary Figure 2 (S2) regression values and p values should be added to each plot

- A Table including the average responses of both emotional (Valence, Arousal, Liking etc) and subjective measures of nutrition (total calories, % carbo, % fat, % protein) should be added to ease the reading of the manuscript

Experimental design

1. The main issue with the experimental design is the unbalanced number of images for Japanese (46 images) vs Western stimuli (5 images), could the authors explain such decision, since this big difference could affect all of the reported statistics.

2. Could the authors explain the small sample size (n=32) since other food images databases have larger samples (FRIDa n = 84; FoodPics n = 638 + n = 831 “german-speaking sample”; OLAF n = 612).

3. The choice of including only processed foods in the study is never explained.

4. How do the authors explain the high ratings on "Naturalness" of Japanese foods given the fact that all of the selected foods were processed foods.

5. (Lines 131-132) The authors do not explain how total calories, percentage of carbohydrate, fat and protein measures were acquired (VAS scale? n-point scale? paper-pencil? on pc?), this should be added to the Procedure section. Giving a look at the raw data a non-continuous variable was used.

6. Did the authors try to analyze the correlation coefficients between the subjective perception of nutrition and emotional responses in participants? Is there a correspondance in the results of such analysis and the reported analysis in Figure 3 (and S1) given the high correlation of objective and subjective nutrition measures (results reported in Figure 4 and S2). If of interest, this further analysis could be added in the Supplementary figures section.

7. Line 105 the image presentation was controlled by Powerpoint 2007 but the authors do not describe how emotional and subjective measures of nutrition were acquired, were the 9-point scales presented on paper-pencil or on the pc? How did the participants report the subjective measures of nutrition (see point 5)?

Validity of the findings

1. The most interesting finding of the research is not stressed by the authors, the fact that objective and subjective measures of nutrition are highly correlated is of great interest (given the findings of Horne et al. 2019, Appetite, which find that most participants were unable to accurately estimate the caloric content of most of the presented foods), here participants are very accurate in estimating the total caloric content (very clear from figure S2), as well as carbohydrate, fat (smaller effect) and protein content.
The authors should stress such finding in the abstract and the discussion, instead of stressing the fact that previous databases did not contain objective nutrition information which is also not completely accurate given the objective values included in Western-food image databases

Additional comments

1. Did the authors acquire information regarding participants’ hunger level at the moment of the experiment? If not, why? given the effects of hunger on explicit ratings (see Coricelli et al. 2019, Food Quality and Preference).

2. Did the authors acquire any information regarding participants’ eating habits? Were the participants all omnivores? Were vegetarians/vegans included in the study? Such info is not present in the manunscript nor in the raw data. If not, why?
Additionally, was information regarding dieting habitudes of participants acquired? given the big amount of literature and the well-known effects of dieting (Restrained eaters) on explicit ratings (see Hoefling & Strack, 2008, Appetite and Coricelli et al. 2019, Food Quality and Preference).

3. Low-level objective measures of the images are not reported in the manuscript (i.e. brightness, spatial frequency, image complexity), why?

Minor changes:

- Line 32 replace the term “better” with another term more specific to the results.

- Line 43 replace the term “controlling eating behaviors” with a more exhaustive explanation of the role of visual processing in food perception such as the ability to extract inherent information of the presented foods (i.e. edibility, Tsourides et al., 2018, NeuroImage).

- Paragraph from 224-233 must be revised; both objective (caloric/fat, healthiness, level of processing) measures and subjective measures (ratings) have been investigated with EEG (see Toepel et al., 2009, NeuroImage; Bielser et al. 2015, Brain and Cognition) and fMRI (see Killgore et al., 2003, NeuroImage; Frank et al., 2010, Brain Research).

---

## Round 0.2 · Minor Revisions

· Academic Editor

Minor Revisions

As you will see below, the reviewers consider your manuscript has clearly improved with the modifications performed. Nevertheless, there are still some minor aspects that should be corrected.

·

Basic reporting

No comment

Experimental design

No comment

Validity of the findings

No comment

Additional comments

The authors did a nice job in revising the manuscript, and appropriately responded to all remarks of the reviewers.

·

Basic reporting

No comment

Experimental design

No comment

Validity of the findings

No comment

Additional comments

Overall there was an improvement in the revised version of the manuscript, in particular in the Introduction section and the paragraph of the Discussion assessing the limitations of the study. The complete set of stimuli in the Supplemental Figure 1 eases the reader to understand the type of food stimuli chosen. The additional results regarding the low-level features (i.e. brightness, sp. frequency, entropy) comprising three Tables of results should be moved to the Supplemental material.
Below specific modifications to the text that should be implemented.

- Line 35 I suggest replacing “the existing database” with “an existing database”

- Line 192 I suggest replacing the term “physical” with “medical”

- Line 198 I suggest replacing the term “visual deficiencies” with the term “color vision deficiencies”

- Line 268 "Participants" should be in lower case

- The authors must make sure they were coherent with the terminology throughout the paper, in particular with the terms “subjective affective ratings” “subjective appraisals” and “subjective evaluations”, (i.e. lines 157-162, 322-326). One term should be chosen and used in the entire manuscript/figures/tables and captions.

- Line 464 “are valid visual stimuli of Japan foods” consider revising this sentence

- Lines 519-521 “Use of the current database may allow studies of the specific brain activities associated with a visual analysis of food nutrition.” This sentence must be reformulated (or removed). Visual processing of food stimuli is quite consistent across studies as shown by the results of meta-analyses and reviews of neuroimaging data (correctly cited in the Introduction), the proposed database contains most of the information contained in other image databases therefore it would not reveal “new” brain activities of visual analysis of food stimuli. It also presents plastic representations of foods, factor that must be taken into account in comparing the visual processing of such stimuli.

- Line 554 I suggest replacing “our stimuli contained different plates” with “our food stimuli were depicted in different plates”

- Line 566-567 “restricted participants” remove “restricted”

- Results related to the low-level features should be moved to the Supplemental material.
I propose to move Table 6, Table 7 and Table 8 to Supplemental material.
Moreover, Table 7 and Table 8 should have a consistent order of statistics (t, p and r or r, t, p)

---

## Round 0.3 · accepted · Accept

· Academic Editor

Accept

The authors have properly addressed all reviewers' comments. I consider the manuscript has improved with the modifications and it is now acceptable for publication.